# Optimization of Longitudinal Alignment of an 4*f* System in a Compact Vectorial Optical-Field Generator Based on a High-Resolution Liquid Crystal Spatial Light Modulator

**Mingyu Li \* and Yuanzheng Liu**

School of Optical-Electrical and Computer Engineering, University of Shanghai for Science and Technology, Shanghai 200093, China; 1815080124@st.usst.edu.cn
\* Correspondence: 2035051415@st.usst.edu.cn

**Abstract:** Vectorial optical fields have garnered significant attention due to their potential applications in areas such as optical nano-fabrication, optical micromachining, quantum information processing, optical imaging, and so on. Traditional compact vectorial optical generators with amplitude modulation perform poorly in terms of diffraction effect reduction. To tackle this problem, the refractive 4*f* system in amplitude modulation is longitudinally aligned using an optimization approach presented in this research. The phase images used for longitudinal alignment are loaded into the liquid crystal spatial light modulator (SLM), and the distance between the lens and the mirror in the reflective 4*f* system is adjusted for longitudinal alignment by compensating for the neglected phase in the integrated module for the compact vectorial optical-field generator. The spot images collected by the CCD are processed using the improved eight-direction Sobel operator and Roberts function, and the longitudinal alignment in the reflective 4*f* system is determined by the sharpness of the image. The sharpness of the edges of the lines and the overall image are both enhanced after optimization compared to before optimization. The results demonstrate that the proposed method can effectively reduce the longitudinal alignment error of the reflective 4*f* system in the amplitude modulation of the compact vectorial optical-field generator, lessen the diffraction effect, and improve the performance of the system.

**Keywords:** vectorial optical-field generator; spatial light modulator; 4*f* system; amplitude modulation; eight-direction Sobel operator

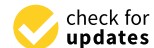



## 1. Introduction

The polarization states of the vectorial optical field [1–5] undergo consistent changes with spatial variations, and a variety of polarization states will appear on the cross section of the optical field [6,7]. Currently, complex vectorial optical fields have found extensive utilization in many fields such as optical micromanipulation technology [8,9], focal field engineering [10,11], quantum information processing [12,13], super-resolution microscopic imaging [14,15], optical trapping and manipulation [16–18], optical communication [19] and laser processing technology [20].

Due to its high resolution and ability to implement real-time control through programming, SLM is frequently employed in the construction of vectorial optical-field generators. How to generate desired beams in complex optical systems has gained significant attention in various related fields. For instance, a versatile vectorial optical-field generator based on two high-resolution reflective SLMs was constructed, and is capable of independently controlling the four degrees of freedom of the vectorial optical field at the pixel level [21]. Simultaneously, the pattern is designed to facilitate the alignment of the 4*f* system of the vectorial optical-field generator. In order to convert linearly polarized beams into vectorial beams with arbitrary spatial polarization and phase distribution, Zhou, Y., proposed a compact optical module capable of efficiently generating vectorial vortex beams [22]. In

order to control the phase, amplitude, polarization ratio and retardation of the optical field in the spatial distribution, Chen, J., designed a compact vectorial optical-field generator utilizing a high-resolution liquid crystal spatial light modulator that can generate arbitrarily complex vectorial beams by controlling the four degrees of freedom of the optical field while effectively minimizing the spatial volume and number of optical elements [23].

4*f* systems are often used in optical information processing fields such as spatial filtering and optical information transmission. In 2005, Pyhtila, J., analyzed the performance of three imaging systems to detect near-forward scattered light interferometrically by using a Mach–Zehnder geometry. Of the three systems analyzed, the 4*f* imaging system was determined to be the most effective because it accurately reproduces both the phase and the amplitude of the scattered field at the detector [24]. In 2015, Wei, Han., proposed a vectorial optical-field generator using a reflective 4*f* system and designed a pattern composed of three horizontal and vertical cross-lines to achieve the longitudinal alignment of the reflective 4*f* system, thus minimizing the diffraction effect. However, the work was to directly observe the effect of longitudinal alignment with a CCD [25]. In 2017, Chen, J., demonstrated a method for the accurate transverse alignment of 4*f* systems in the vectorial optical-field light-field generator. Based on the Fourier optics approach, the coordinate relationships among four sections of SLM in the vectorial optical-field generator were derived and experimentally verified [26].

In this work, the compact vectorial optical-field generator utilizing a high-resolution liquid crystal spatial light modulator includes two 4*f* systems. The reflective 4*f* system serves to relay the beam to the corresponding region of SLM for modulation, and the transmitted 4*f* system transmits the output optical field to the CCD for data collection. In reference [23], the longitudinal alignment of the reflective 4*f* system has a notable inaccuracy in the amplitude modulation for the compact vectorial optical-field generator. To tackle this problem, lessen the diffraction effect of the beam during propagation and make the obtained output optical field as clear and sharp as possible, it is essential to optimize the longitudinal alignment of the reflective 4*f* system. By compensating for the neglected phase in the integrated module of the compact vectorial optical-field generator, the phase images used for longitudinal alignment are loaded into the SLM, and the distance from lens to mirror in the reflective 4*f* system is adjusted for longitudinal alignment.

## 2. Experimental Device and Principle

### 2.1. Compact Vectorial Optical-Field Generator

The experimental device of the compact vectorial optical-field generator is shown in Figure 1, which is primarily composed of two 4*f* systems, the integrated module, CCD, and SLM. The reflective 4*f* system encompasses the large lens and the mirror, while the transmitted 4*f* system comprises lenses 1 and 2. The integrated module, as depicted in Figure 2, incorporates a rectangular prism, one square half-wave plate (HWP), two square quarter-wave plates (QWPs), one polarizing beam splitter (PBS) and four non-polarizing beam splitters (NPBSs).

The fast axes of all three plates were positioned at 45 degrees with respect to the horizontal direction. In order to avoid mutual crosstalk caused by the transmitted and reflected beams, the overlapping surface between NPBS 1 and NPBS 3 was appropriately blackened. Similarly, the surface of PBS adjacent to NPBS 4 was also blackened for the same purpose. The collimated horizontally polarized He–Ne laser with a wavelength of 632.8 nm and a diameter of 3 mm was utilized as the source. The SLM with 4K resolution was subdivided into four distinct sections that could independently and simultaneously modulate the four degrees of freedom of the vector beam. Figure 3 illustrates the locations of the beam to be modulated on each SLM section, which were determined by the integrated module. The phase, amplitude, polarization ratio and retardation were modulated, respectively, in Section 1–4.

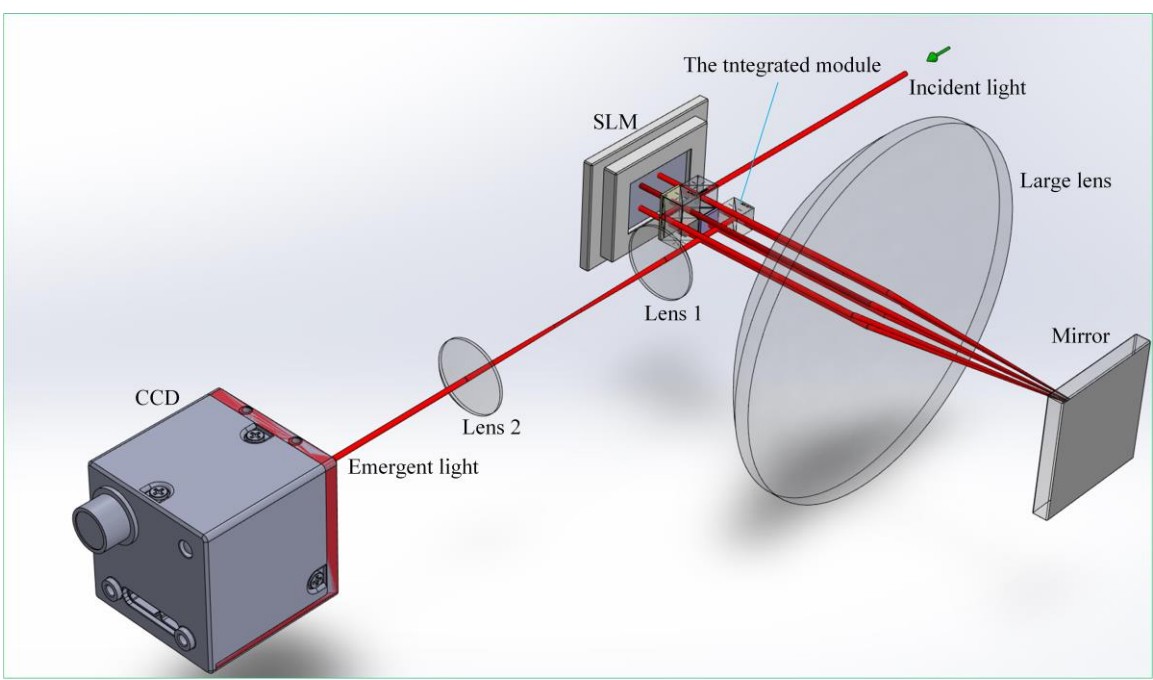

**Figure 1.** The experimental device of the compact vectorial optical-field generator.

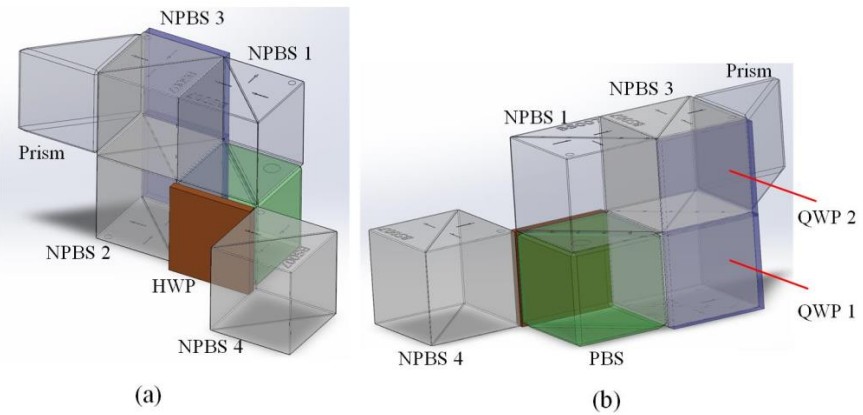

**Figure 2.** (**a**) Front view of the integrated module; (**b**) rear view of the integrated module.

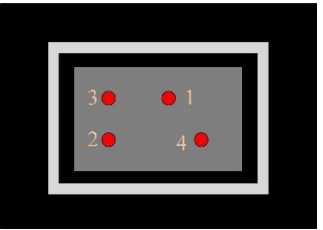

**Figure 3.** The locations of the beam to be modulated on each SLM section.

The optical path in the integrated module is depicted in Figure 4. Initially, the incident beam was steered by NPBS 1 towards Section 1 of the SLM for phase modulation. Subsequently, the reflected beam was relayed through the reflective 4*f* system towards NPBS 2 and transmitted back to Section 2 of SLM. The SLM is a phase modulator, so the amplitude modulation was accomplished by employing a combination of the polarizing rotator and PBS. The polarizing rotator is composed of QWP and Section 2 of the SLM, and the specific principle can be found in reference [20]. The modulated beam reflected by Section 2 of

the SLM was guided from NPBS 2 to PBS with the optical axis in the vertical direction. Subsequently, the HWP changed the polarization of the beam to the horizontal direction. The beam was relayed by the same reflective 4*f* system to NPBS 3 and transmitted to the rear Section 3 of the SLM for polarization ratio modulation. In this case, QWP and Section 3 of the SLM constituted another polarizing rotator. The beam was reflected from Section 3 of the SLM through NPBS 4 to the rectangular prism and relayed to NPBS 4 by the reflective 4*f* system, where the beam was modulated by the retardation after passing through Section 4 of the SLM. At this time, the integrated module depicted in Figure 2 effectively achieved the modulation of 4 degrees of freedom in the vectorial optical field. Finally, the beam was relayed to the CCD by a transmitted 4*f* system. In order to generate an accurate vectorial optical field, the diffraction effect of the beam propagation must be reduced by the 4*f* system, which requires the longitudinal alignment of the 4*f* system [23].

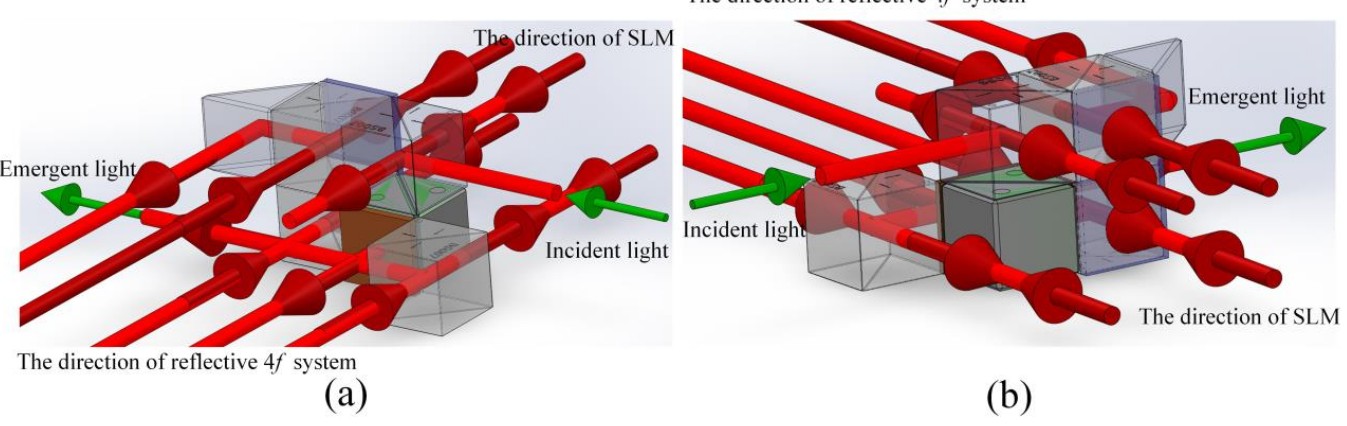

**Figure 4.** (**a**) Front view of the optical path in the integrated module; (**b**) rear view of the optical path in the integrated module.

### *2.2. 4f System*

#### 2.2.1. The Structure of the 4*f* System

The beam can be transformed by the 4*f* system from the spatial domain to the frequency domain and then back to the spatial domain. The compact vectorial optical-field generator comprises two 4*f* systems. One of them is the transmitted 4*f* system, as illustrated in Figure 5a. The transmitted 4*f* system comprises two lenses, both of which have a focal length of 20 cm. The other is a reflective 4*f* system, as depicted in Figure 5b. The reflective 4*f* system consists of a lens with a focal length of 10 cm and a mirror.

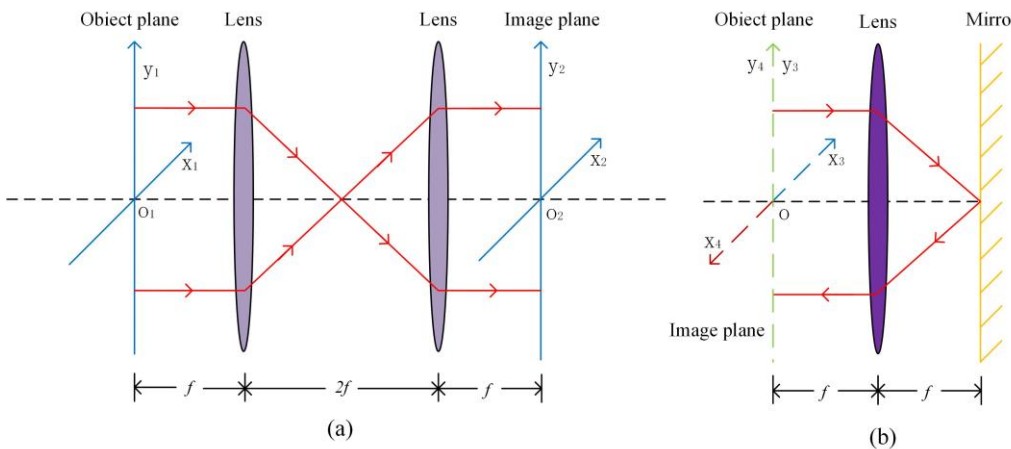

**Figure 5.** (**a**) The transmitted 4*f* system; (**b**) the reflective 4*f* system.

### 2.2.2. The Longitudinal Alignment of the 4*f* System

Initially, the shear interferometer is employed to determine whether the distance between lens and mirror is *f* [27]. It is generally possible to estimate if the distance between them has met the requirements when the highest interference fringe on the shear interferometer is observed to be parallel to the line of reference. When the distance of the object plane and the image plane to the lens both are *f*, the longitudinal alignment of the 4*f* system is theoretically deemed complete. However, the distance between the lens and mirror in the reflective 4*f* system is no longer an ideal distance due to the optical path differences and experimental errors; the reflective 4*f* systems need to be slightly regulated. This includes the longitudinal alignment of the reflective 4*f* system for amplitude modulation. When the amplitude modulation region is longitudinally aligned, a specially aligned pattern is adopted. The corresponding aligned pattern is loaded in the amplitude modulation region of SLM and the position of the large lens and the mirror in the reflective 4*f* system is adjusted by the micrometer on the experimental platform. The data are collected by CCD after longitudinal alignment.

### 2.3. The Amplitude Modulation Principle of the Compact Vectorial Optical-field Generator

The first is phase modulation. Since SLM is a device with an external phase delay, the phase of the optical field can be directly adjusted. According to reference [28], the optical field after phase modulation can be accurately described by the Jones matrix as the following formula.

$$J_1(x,y) = e^{j\varphi_1(x,y)} E_i(x,y) \begin{bmatrix} 1 \\ 0 \end{bmatrix}, \tag{1}$$

In Formula (1), $E_i(x,y)$ is the amplitude intensity distribution of the incident optical field and $\varphi_1(x,y)$ is the phase information loaded into Section 1 of the SLM.

The second is amplitude modulation. The polarizing direction of the optical field spins at a specific angle after the incident beam passes through the polarizing rotator before entering PBS. According to Malus' law, the amplitude of the optical field is modified by PBS because of the angle between the polarizing direction of the optical field and the optical axis of PBS. The vertically polarizing beam generated by PBS is converted into a horizontally polarizing beam. The optical field after amplitude modulation can be represented by the Jones matrices as the following formula.

$$J_2(x,y) = M_{HWP} M_s M_{PR}(\varphi_2(x,y)) J_1(x,y)$$
$$= e^{j[\varphi_1(x,y) + \frac{\varphi_2(x,y)}{2} + \frac{\pi}{2}]} E_i(x,y) \cos \frac{\varphi_2(x,y)}{2} \begin{bmatrix} 1 \\ 0 \end{bmatrix}, \tag{2}$$

In Formula (2), $\varphi_2(x,y)$ is the phase information loaded into Section 2 of SLM. $M_{HWP}$, $M_s$ and $M_{PR}(\varphi_2(x,y))$ are the Jones matrices for HWP, the reflection of PBS and the polarizing rotator, respectively, which can be given as the following:

$$M_{HWP} = j \begin{bmatrix} 0 & 1 \\ 1 & 0 \end{bmatrix}, \ M_s = \begin{bmatrix} 0 & 0 \\ 0 & 1 \end{bmatrix},$$
$$M_{PR}(\varphi_2(x,y)) = e^{j\frac{\varphi_2(x,y)}{2}} \begin{bmatrix} -\sin \frac{\varphi_2(x,y)}{2} & -\cos \frac{\varphi_2(x,y)}{2} \\ \cos \frac{\varphi_2(x,y)}{2} & -\sin \frac{\varphi_2(x,y)}{2} \end{bmatrix}. \tag{3}$$

In fact, the beam exists a distance from NPBS 2 to PBS in the integrated module during amplitude modulation and the distance is approximately 5 mm in length for a polarization beam splitter. However, according to reference [23], when the reflective 4*f* system is longitudinally aligned during amplitude modulation, it is considered that the distance of the beam reflected from Section 2 of SLM to the large lens is the same as that from the large lens to Section 3 of SLM. Therefore, the optical path generated by the transmission of the beam in the integrated module is neglected. In order to ensure the effectiveness of the Fourier transform of the 4*f* system, it is necessary to ensure that the

distance from the object plane to the lens is equal to the focal length. At this moment, the Fourier transform of the input light in the frequency domain corresponds to the optical-field distribution in the Fourier plane. However, the neglected optical path leads to the fact that the distance from the object plane (Section 2 of SLM) of the reflective $4f$ system to the large lens cannot be strictly defined as the focal length in the amplitude-modulated optical path. According to Fourier optics, when the phase delay generated by the optical field in the propagation process is an integer multiple of $2\pi$, the optical density distribution on the Fourier plane will not be affected by the phase. In order to obtain a better sharpness of the image observed by CCD, the phase modulation region on the SLM facilitates compensation for the neglected phase that is $\Delta\varphi$ during longitudinal alignment of the reflective $4f$ system in amplitude modulation. The sum of the compensated phase and the phase generated by the transmission in the integrated module reaches $2\pi$, which can reduce the influence of this optical path on the reflective $4f$ system. The compensated phase can be calculated as the following:

$$\varphi = \frac{2\pi}{\lambda}\Delta x = \varphi' + 2n\pi, n = 0, 1, 2 \cdots$$
$$\Delta\varphi = 2\pi - \varphi' \tag{4}$$

In Formula (4), $\Delta x$ is the transmission distance of the beam from NPBS 2 to PBS, $\lambda$ is the wavelength and the phase delay that actually affects the $4f$ system is $\varphi'$. By calculation, the compensated phase should be $1.2188\pi$.

### 2.4. The Sharp Evaluation of the Image

### 2.4.1. The Eight-Direction Sobel Operator

In order to obtain a more accurate observation of the sharpness of the image after longitudinal alignment of the reflective $4f$ system in amplitude modulation, this paper uses an improved eight-direction Sobel operator to extract the contour of the image and evaluate the sharpness of the image by observing the edge of the contour. Compared with the traditional Sobel operator, the image contour is more accurate. Using the Sobel operator to detect the edge of the image is a complex process, but it is relatively less affected by noise and has a higher accuracy.

Compared with the traditional Sobel operator, the eight-direction Sobel operator is increased to eight directions based on the only horizontal and vertical directions of the template. The eight-direction Sobel operator proposed in reference [29] uses a $5 \times 5$ directional template. Although the edge of the image is better detected, the influence of the noise is neglected, which worsens the overall detection effect, increases the overall calculation, and slows down the detection speed. In order to address this issue, this paper adopts a $3 \times 3$ directional template. The templates for each direction are as follows:

$$g_1 = \begin{bmatrix} -1 & 0 & 1 \\ -2 & 0 & 2 \\ -1 & 0 & 1 \end{bmatrix}, g_2 = \begin{bmatrix} -2 & -1 & 0 \\ -1 & 0 & 1 \\ 0 & 0 & 2 \end{bmatrix}, g_3 = \begin{bmatrix} -1 & -2 & -1 \\ 0 & 0 & 0 \\ 1 & 2 & 1 \end{bmatrix}, g_4 = \begin{bmatrix} -1 & 0 & 1 \\ -2 & 0 & 2 \\ -1 & 0 & 1 \end{bmatrix},$$
$$g_5 = \begin{bmatrix} 1 & 0 & -1 \\ 2 & 0 & -2 \\ 1 & 0 & -1 \end{bmatrix}, g_6 = \begin{bmatrix} 2 & 1 & 0 \\ 1 & 0 & -1 \\ 0 & -1 & -2 \end{bmatrix}, g_7 = \begin{bmatrix} 1 & 2 & 1 \\ 0 & 0 & 0 \\ -1 & -2 & -1 \end{bmatrix}, g_8 = \begin{bmatrix} 0 & 1 & 2 \\ -1 & 0 & 1 \\ -2 & -1 & 0 \end{bmatrix}. \tag{5}$$

In Formula (5), $g_1, g_2, g_3, g_4, g_5, g_6, g_7$ and $g_8$, respectively, represent templates in the directions of $0°, 45°, 90°, 135°, 180°, 225°, 270°$ and $315°$.

Firstly, the image to be detected is convolved with eight directional templates to obtain the information about the contour of the image, which is $G_1, G_2, G_3, G_4, G_5, G_6, G_7$ and $G_8$. The gray value at each pixel point can be calculated as the following:

$$G_a = G_1{}^2 + G_2{}^2 + G_3{}^2 + G_4{}^2, \tag{6}$$

$$G_b = G_5{}^2 + G_6{}^2 + G_7{}^2 + G_8{}^2, \tag{7}$$

$$G'(x,y) = \frac{1}{2}\sqrt{G_a + G_b}.$$ (8)

In Formula (8), $G'(x, y)$ is the gray value at each pixel point. In order to improve the efficiency of operation, this paper used Formula (9) instead of Formula (8) to reduce the amount of calculation. Finally, a suitable threshold value ($T$) was selected and the gray value of each pixel was performed as a threshold operation. If the gray value of a pixel is greater than or equal to $T$, it is determined to be a contour point; otherwise, it is determined to be a non-contour point. The formulae for extracting the contour of the image are as follows:

$$G'(x,y) = \frac{1}{2}(|G_1| + |G_2| + |G_3| + |G_4|),$$ (9)

$$E(x,y) = \begin{cases} 0, & G'(x,y) < T \\ 255, & G'(x,y) \geq T \end{cases}.$$ (10)

In Formula (10), $E(x, y)$ represents the gray value of the pixel after the threshold operation.

### 2.4.2. The Roberts Function

In this paper, the Roberts function was used to quantitatively evaluate the improvement in phase-compensated optical spot image quality. The optical image was regarded as a two-dimensional discrete matrix, and the gradient was represented as a difference in the discrete signal. The Roberts function was used to obtain the gray information of the optical image to judge the sharpness of the image. Using the difference of the gray values of the diagonal pixels, the Roberts function takes the sum of the squares of the cross subtraction of the gray values of the four adjacent pixels as the gradient value of each pixel. The value of the sharpness evaluation function is the sum of the gradient values for each pixel [30]. The formula for the Roberts function is as follows:

$$F = \sum_x \sum_y \left\{ [f(x+1, y+1) - f(x,y)]^2 + [f(x+1) - f(x,y+1)]^2 \right\}.$$ (11)

In Formula (11), $f(x, y)$ represents the gray value of the pixel on the optical spot image.

### 3. Experimental Results and Discussion

A specially aligned pattern was used for longitudinal alignment of the reflective 4*f* system in amplitude modulation. Figure 6 is the gray image loaded into SLM during longitudinal alignment. The pattern is composed of three pairs of horizontally and vertically crossed lines. When the reflective 4*f* system was longitudinally aligned in amplitude modulation, it was set to pass all light on the lines and suppress all light outside the lines. The final image is in an upside-down relationship with the object, so the four sections designed in the gray image were in an upside-down relationship with the four sections of the SLM shown in Figure 3. There are four square sections in the gray image. According to the optical path in the integrated module, the square section corresponding to the upper left corner corresponds to Section 2 of the SLM, which is the amplitude modulation area; the square section in the lower left corner corresponds to Section 3 of the SLM, which is the polarization ratio modulation area; the square section in the upper right corner corresponds to Section 4 of the SLM, which is the phase difference modulation area; and the square section in the lower right corner corresponds to Section 1 of the SLM, which is the phase modulation area. The workflow of the compact vectorial optical-field generator was the phase modulation, amplitude modulation, polarization ratio modulation, and phase difference modulation. When the reflective 4*f* system was longitudinally aligned in amplitude modulation, the neglected phase was compensated for in Section 1 of the SLM. Figure 6a is the gray image of the uncompensated phase; Figure 6b is the gray image of the 1/4-compensated phase; Figure 6c is the gray image of the 1/2-compensated phase;

Figure 6d is the gray image of the 3/4-compensated phase; and Figure 6e is the gray image of the fully compensated phase. The final optical field was collected by the CCD [26].

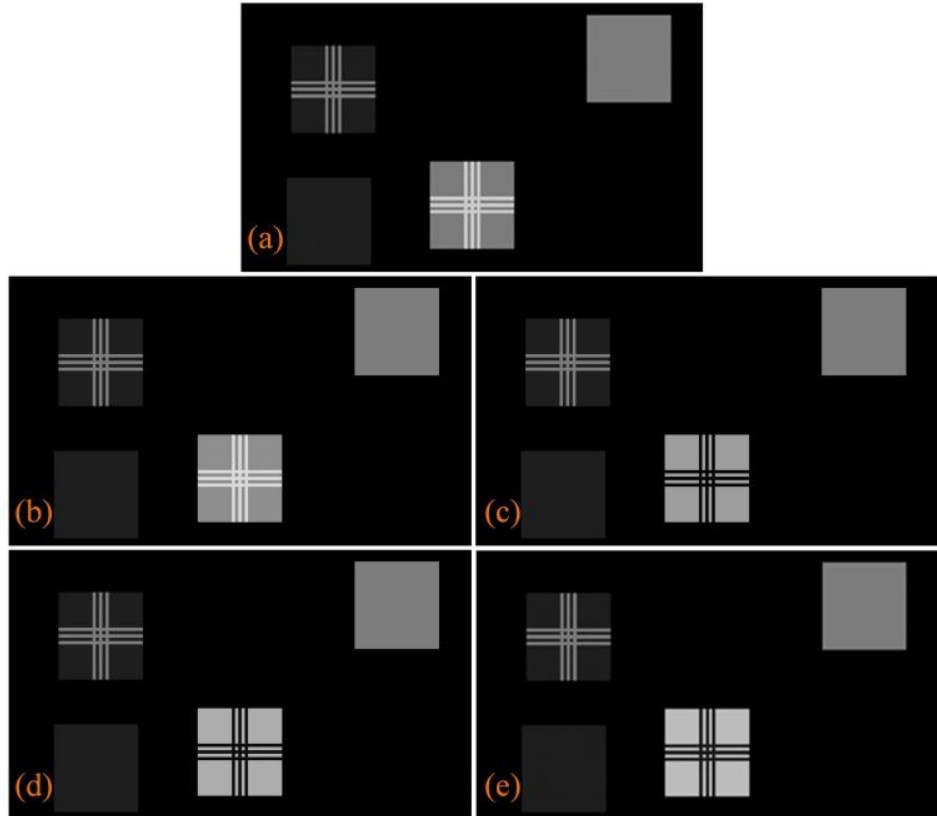

**Figure 6.** Longitudinally aligned gray images in amplitude modulation. (**a**) The uncompensated phase; (**b**) the 1/4 compensated phase; (**c**) the 1/2 compensated phase; (**d**) the 3/4 compensated phase; and (**e**) the fully compensated phase.

The optical spot image with the uncompensated phase collected by the CCD is shown in Figure 7a; the optical spot image with the 1/4-compensated phase is shown in Figure 7b; the optical spot image with the 1/2-compensated phase is shown in Figure 7c; the optical spot image with the 3/4-compensated phase is shown in Figure 7d; and the image of the optical spot with the compensated phase is shown in Figure 7e, which correspond to the gray images of Figure 6a–e, respectively. When the spot images with different phase compensation degrees are collected, the light energy values received by the CCD are shown in Table 1. The unit of the light energy value is *cnts*. The improved eight-direction Sobel operator was used to process five images of the optical spot to obtain corresponding contours, as shown in Figure 8a–e. The Roberts function is used to process the optical spot images in Figure 7a–e, and corresponding sharpness values are obtained, respectively, as shown in Table 2. With the increase in phase compensation degree, the sharpness of the optical spot images and the sharpness of the contour images increase. Compared with the contour without the compensated phase, the edge of the contour of the three horizontal lines became significantly sharper, and the sharpness of the overall image contour was improved. The diffraction effect of the optical field was reduced, and the degree of longitudinal alignment of the reflective 4*f* system in amplitude modulation was improved.

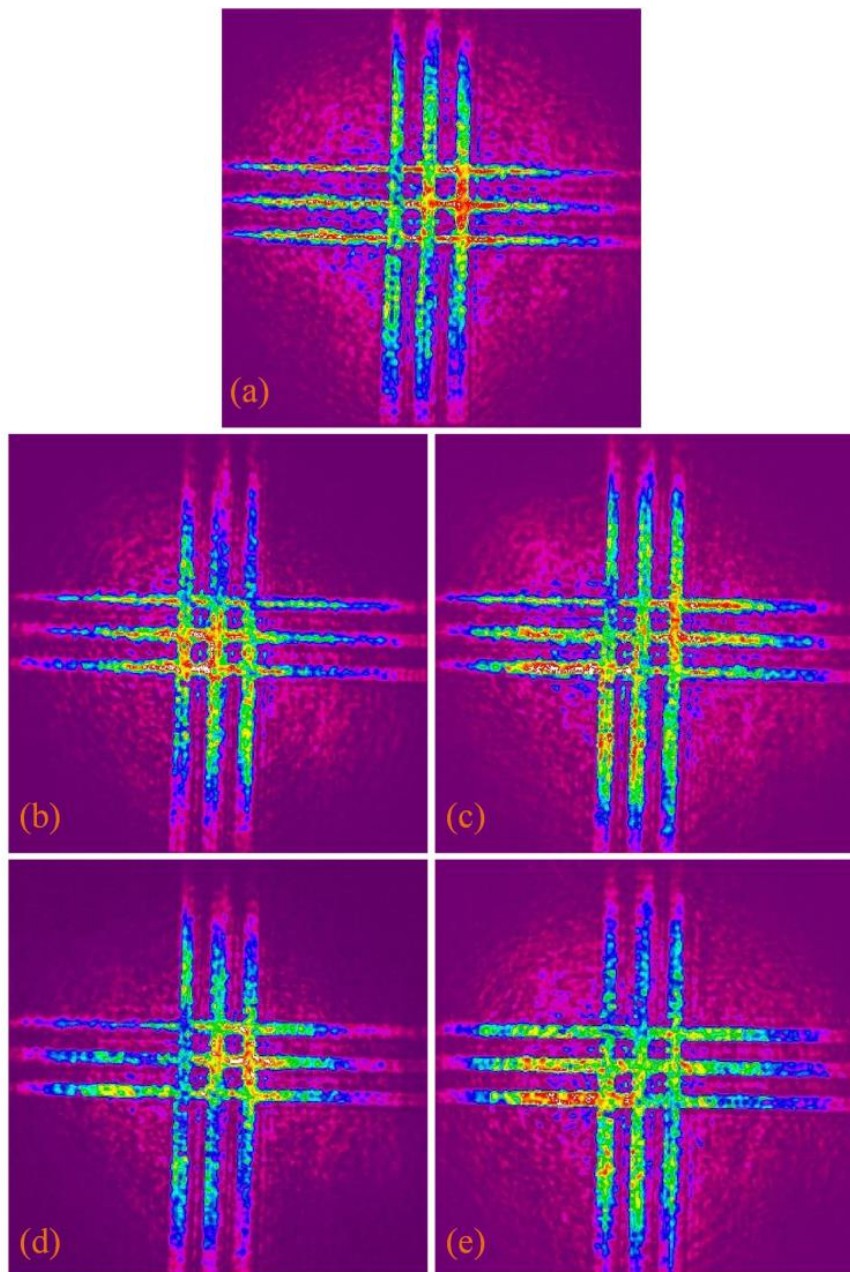

**Figure 7.** (**a**) The image of optical spot with the uncompensated phase; (**b**) the image of optical spot with the 1/4-compensated phase; (**c**) the image of optical spot with the 1/2-compensated phase; (**d**) the image of optical spot with the 3/4-compensated phase; and (**e**) the image of the optical spot with the fully compensated phase.

**Table 1.** The light-energy values of the optical spot images.

| The Optical Spot Images | The Light Energy Values |
| --- | --- |
| 0 | 724,034,311.19 |
| 1/4 | 697,780,583.14 |
| 1/2 | 826,344,804.78 |
| 3/4 | 633,813,389.00 |
| 1 | 686,753,913.44 |

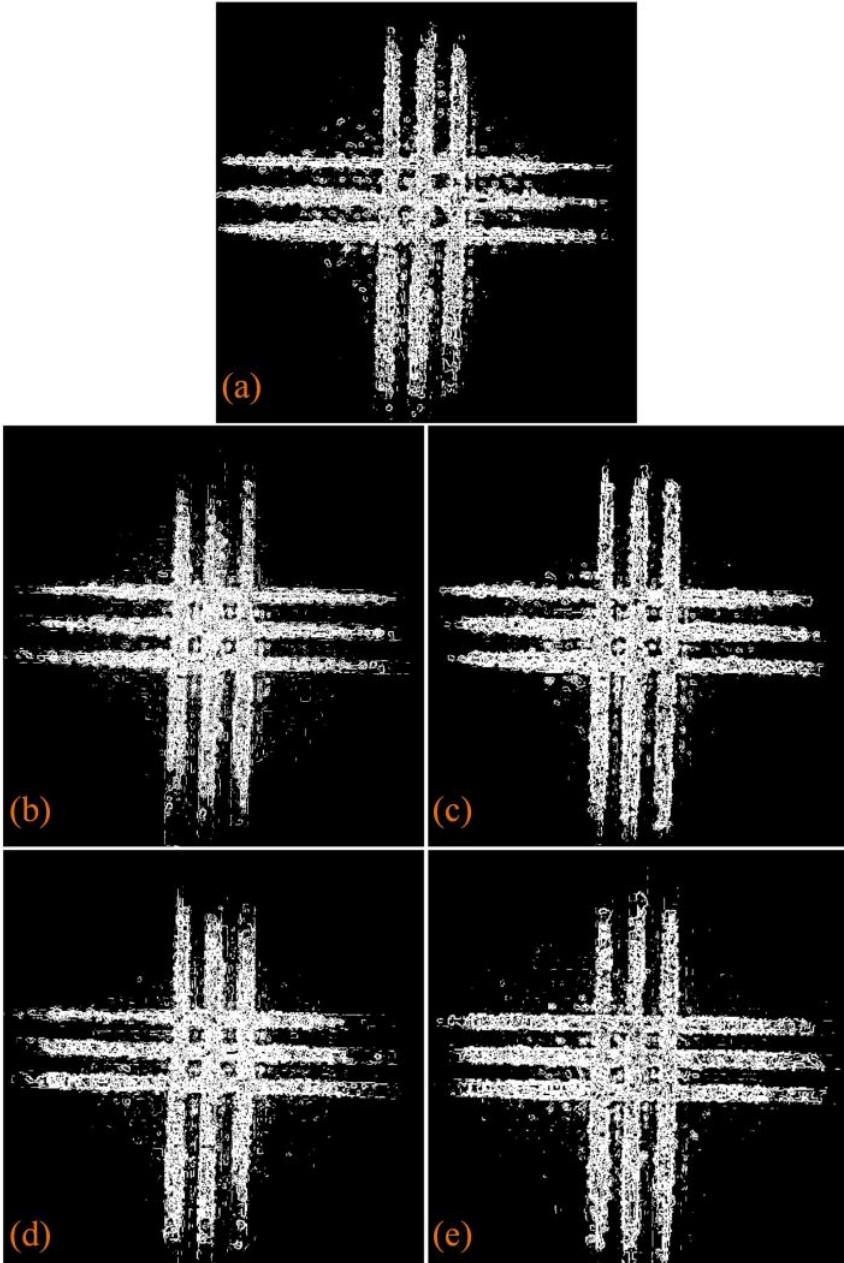

**Figure 8.** (**a**) Extract of the contour from the optical spot image with the uncompensated phase; (**b**) extract of the contour from the optical spot image with the 1/4-compensated phase; (**c**) extract of the contour from the optical spot image with the 1/2-compensated phase; (**d**) extract of the contour from the optical spot image with the 3/4-compensated phase; and (**e**) extract of the contour from the optical spot image with the fully compensated phase.

**Table 2.** The sharpness values of the optical spot images.

| The Optical Spot Images | The Sharpness Values ($\times 10^7$) |
| --- | --- |
| 0 | 1.5056499 |
| 1/4 | 1.7762937 |
| 1/2 | 2.1146577 |
| 3/4 | 2.3256987 |
| 1 | 2.5117032 |

## 4. Conclusions

This paper presents an optimization of the method for the longitudinal alignment of the reflective 4*f* system in amplitude modulation of a compact vectorial optical-field generator based on SLM. In this paper, the composition of the compact vectorial optical-field generator, the optical path in the integrated module, the structure of 4*f* system, the principle of amplitude modulation in the compact vectorial optical-field generator, the improved eight-direction Sobel operator and the Roberts function were introduced. By compensating the neglected phase to optimize the reflective 4*f* system and using the improved eight-direction Sobel operator and the Roberts function to process the optical spot images, the effect of longitudinal alignment can be more accurately judged. The experimental results show that compared with the contour without compensated phase, the edge of the contour of the three horizontal lines becomes significantly sharper and the sharpness of the overall image contour is improved. The diffraction effect of the optical field is reduced. The longitudinal alignment of the reflective 4*f* system is optimized in the amplitude modulation of the compact vectorial optical-field generator. The amplitude modulation is a necessary process to build a vectorial optical-field generator and generate complex vectorial optical fields. This method can improve the performance of the vectorial optical-field generator.

**Author Contributions:** Conceptualization, M.L. and Y.L.; methodology, M.L. and Y.L.; software, M.L. and Y.L.; validation, M.L. and Y.L.; formal analysis, M.L. and Y.L.; investigation, Y.L.; resources, Y.L.; data curation, M.L.; writing—original draft preparation, M.L. and Y.L.; writing—review and editing, M.L. and Y.L.; visualization, M.L. and Y.L.; supervision, M.L. and Y.L.; project administration, M.L. and Y.L.; funding acquisition, M.L. All authors have read and agreed to the published version of the manuscript.

**Funding:** This research received no external funding.

**Institutional Review Board Statement:** Not applicable.

**Informed Consent Statement:** Not applicable.

**Data Availability Statement:** Data will be made available on request.

**Conflicts of Interest:** The authors declare no conflict of interest.

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
