# Peer review of "Optimization of Longitudinal Alignment of an 4f System in a Compact Vectorial Optical-Field Generator Based on a High-Resolution Liquid Crystal Spatial Light Modulator"

_photonics, doi:10.3390/photonics10080894_

Round 1

Reviewer 1 Report

     In this paper, the authors optimize the 4f systems by compensating for neglected phases, thus reducing diffraction efficiency as well as improving graphics quality. The work contains a description of the optical system and principle of operation as well as results of imaging comparisons.  In addition, the phase compensation is an important part of the article and the writer needs to provide more relevant content.

     1:This article is about optimizing image quality through alignment matching techniques. However, in the introduction, the authors spend a lot of time describing the vector light field, and there is a lack of literature tracking on diffraction efficiency and alignment optimization techniques for 4f systems. It would be useful to have a comparable table in chronological order.

     2:There is an improvement in the image profile in Fig. 7, but it may be not very noticeable. Can the authors use a formula or data to quantitatively evaluate the improvement in image quality?

     3: The authors have shown images in the uncompensated phase and compensated phase states. Can authors add some images under small partially compensated, half compensated and mostly compensated states. This will help us to see the whole process of phase compensation to enhance the image quality.

    4:Could the authors please show the values of light energy received by the CCD camera under different phase compensation conditions.

Fine changes are required

Reviewer 2 Report

The paper describes fine tuning of a vectorial optical field generator to optimise its performance. The proposed use of the eight-direction Sobel operator makes the effect of the phase compensation alignment more visible and, as a result, the quality of the alignment more efficient. Overall, the research is a well-balanced combination of theory and experiment, all the required details are included. A number of corrections would be good before publishing:

The possessive form like lines' edges (21), vectorial beam’s (82) is not really good for formal scientific writing.

52 for date collection  - probably you mean data collection.

60 it is necessary to optimize the longitudinal alignment  - would be good to add one more sentence to say how it is done in the paper.

197 uses the formula (9) instead of the formula (8)  - why this is possible?

200 greater than or equal to 255  - it looks like it should be T instead of 255 here, according to formula 10.

210 is composed of three pairs of…  - the gray image loaded into SLM has to be described in more details, in addition to the three-line pattern. It has four parts, probably matching the four beam positions on figure 3. What are these parts? Why they are like they are?

232 paper presents an optimal method  - an optimised method or an optimisation of the method would be better.

243 effect of the optical field is lessened  - reduced would be better here.

Included in the previous field
